# Lysosomal ATP Transporter SLC17A9 Controls Cell Viability via Regulating Cathepsin D

**DOI:** 10.3390/cells11050887

**Published:** 2022-03-04

**Authors:** Peng Huang, Qi Cao, Mengnan Xu, Xian-Ping Dong

**Affiliations:** 1School of Clinical Medicine, Shanghai University of Medicine and Health Sciences, Shanghai 201318, China; huangp_15@sumhs.edu.cn; 2Collaborative Innovation Center for Biomedicine, Shanghai University of Medicine and Health Sciences, Shanghai 201318, China; 3Department of Physiology and Biophysics, Dalhousie University, Sir Charles Tupper Medical Building, 5850 College Street, Halifax, NS B3H 4R2, Canada; rachellecao@msl.ubc.ca (Q.C.); mn863235@dal.ca (M.X.)

**Keywords:** ATP transporter, vesicular nucleotide transporter (VNUT), solute carrier family 17 member 9 (SLC17A9), lysosome

## Abstract

SLC17A9 (solute carrier family 17 member 9) functions as an ATP transporter in lysosomes as well as other secretory vesicles. SLC17A9 inhibition or silence leads to cell death. However, the molecular mechanisms causing cell death are unclear. In this study, we report that cell death induced by SLC17A9 deficiency is rescued by the transcription factor EB (TFEB), a master gene for lysosomal protein expression, suggesting that SLC17A9 deficiency may be the main cause of lysosome dysfunction, subsequently leading to cell death. Interestingly, Cathepsin D, a lysosomal aspartic protease, is inhibited by SLC17A9 deficiency. Heterologous expression of Cathepsin D successfully rescues lysosomal dysfunction and cell death induced by SLC17A9 deficiency. On the other hand, the activity of Cathepsin B, a lysosomal cysteine protease, is not altered by SLC17A9 deficiency, and Cathepsin B overexpression does not rescue lysosomal dysfunction and cell death induced by SLC17A9 deficiency. Our data suggest that lysosomal ATP and SLC17A9 play critical roles in lysosomal function and cell viability by regulating Cathepsin D activity.

## 1. Introduction

The solute carrier (SLC) proteins form a large family of membrane transporters located in both the plasma membrane and intracellular membranes, where they play important roles in the transport of a variety of solutes across cell membranes [1,2]. The most recently described family member, SLC17A9, is widely expressed throughout the body [3,4] and represents a vesicular nucleotide transporter (VNUT) [5]. Increasing evidence indicates that SLC17A9 is essential for vesicular ATP accumulation and plays important roles in various systems [3,4]. Recent studies also suggest that lysosomes are ATP stores in many cell types [6,7,8] and SLC17A9 acts as an ATP transporter responsible for the accumulation of ATP in lysosomes [9,10]. SLC17A9 deficiency leads to compromised lysosomes, subsequently causing lysosomal storage and cell death [10]. However, the molecular mechanisms underlying cell death induced by SLC17A9 deficiency remain largely unclear.

Lysosomes have long been known to be key players in the degradation of different substrates in the cell. Dysfunction of lysosomal enzymes promotes lysosome abnormalities and subsequently leads to cell death [11]. Cathepsin D is one of the most extensively characterized enzymes in lysosomes. Cathepsin D deficiency has been associated with lysosomal storage and cell death in mouse models and with human disorders such as neuronal ceroid lipofuscinoses and neurodegeneration [12,13,14,15]. Since SLC17A9 deficiency shares the same phenotypes with Cathepsin D mutation, including lysosomal dysfunction, lysosomal storage, and cell death, we hypothesized that the loss of Cathepsin D activity might be associated with cell death induced by SLC17A9 deficiency. In support of this, previous biochemical studies have suggested that ATP activated Cathepsin D increases proteolysis [16].

The transcription factor EB (TFEB) is a master regulator of lysosome biogenesis. It induces the expression of various lysosomal proteins by upregulating a family of genes belonging to the Coordinated Lysosomal Expression and Regulation (CLEAR) network [17,18,19]. Mounting evidence suggests that upregulation of TFEB increases lysosomal proteolysis, promotes lysosomal storage clearance, and protects cells from damage in lysosomal storage diseases [20,21,22,23,24,25,26,27].

In this study, we showed that cells with compromised SLC17A9 displayed reduced Cathepsin D activity and increased apoptosis. However, Cathepsin B activity was not affected by SLC17A9 deficiency. Importantly, reduced Cathepsin D activity and increased apoptosis in SLC17A9 deficient cells were rescued by expressing TFEB and Cathepsin D but not Cathepsin B. Our data suggest that dysfunction of Cathepsin D might be one major cause of cell death in SLC17A9 deficient cells.

## 2. Materials and Methods

### 2.1. Cell Culture and Transfection

Cos1, Hela and C2C12 cells were used in this study to verify the general role of SLC17A9 in lysosomal functions and cell viability. Cos1, Hela and C2C12 cells were obtained from ATCC (Manassas, VA, USA) and maintained in Dulbecco’s Modified Eagle Medium: Nutrient Mixture F-12 (DMEM/F12) supplemented with 10% fetal bovine serum (Invitrogen, Carlsbad, CA, USA). Cells were cultured at 37 °C in a 5% CO_2_ atmosphere. For some experiments, cells were seeded on 0.01% poly-lysine coated coverslips and cultured for 24 h before further experiments. Cos1 cells were used for DIDS treatment experiments because they had been used to demonstrate that SLC17A9 acted as a lysosomal ATP transporter in our previous studies and because they were easily transfected using Lipofectamine 2000. The mouse myoblast C2C12 cells were used for SLC17A9 shRNA experiments because the pSUPER-SLC17A9 shRNAs we have previously generated are against mouse SLC17A9. C2C12 cells were transfected by electroporation with the Neon^®^ Transfection System (Thermo Fisher Scientific, Waltham, MA, USA) following the manufacturer’s optimized protocol (1650 V, 30 ms, 3 pulses). An ~80% transfection efficiency was regularly achieved. Because both green and red channels are required to determine the modes of cell death using the Apoptosis/Necrosis Assay Kit (ab176749, Abcam, Cambridge, MA, USA), C2C12 cells transfected with pSUPER-mSLC17A9 shRNAs (green) were not compatible with the kit. We established SLC17A9 stable knockdown Hela cells for the apoptosis/necrosis studies.

### 2.2. Antibodies and Reagents

The following primary antibodies were used in immunofluorescent staining and western blotting: anti-Lamp-1 (1D4B, H4A3, Developmental Studies Hybridoma Bank, Iowa City, IA, USA), anti-Cathepsin D (C-20, sc6486, Santa Cruz Biotechnology, Dallas, TX, USA) anti-Cathepsin B (T-12, sc86313) and anti-GAPDH (H-12, sc166574), anti-TFEB (Proteintech, 13372-1-AP, Rosemont, IL, USA). HRP conjugated secondary antibodies were purchased from Santa Cruz.

Texas Red 10 kD Dextran (Invitrogen, 1 mg/mL) was used to label lysosomes. LysoTracker Red DND-99 (Invitrogen, 50 nM) was used to indicate acidic lysosomes. DQ-BSA (Invitrogen, 10 μg/mL) was used to detect lysosomal function. BODIPY FL-pepstatin A (Invitrogen, 1 μM) was used to stain active Cathepsin D. A Magic Red™ Cathepsin B Assay Kit (ImmunoChemistry Technologies, Bloomington, MN, USA) was used to stain active Cathepsin B. 4, 4′-Diisothiocyano-2, 2′-stilbenedisulfonic acid (DIDS, 5 μM, Tocris Bioscience, Bristol, UK) was used to inhibit SLC17A9.

### 2.3. Immunocytochemistry

Cells were loaded with DQ-BSA (10 μg/mL) for 3 h followed by a 1 h chase. DQ-BSA fluorescence was measured to evaluate lysosome function. For staining of active Cathepsin D, cells were incubated at 37 °C for 30 min with a culture medium containing 1 μM BODIPY FL-pepstatin A and rinsed twice with PBS before being viewed. For active Cathepsin B detection, a Magic Red™ Cathepsin B Assay Kit was used according to the manufacturer’s manual. De-staining of DQ-BSA and active Cathepsins was induced by treating cells with 5 μM DIDS for 6 h. After washing with PBS to remove dead cells, images were acquired with a confocal microscope (LSM510, Zeiss, New York, NY, USA) using a 63× oil-immersion objective lens and captured with ZEN2009 software (Zeiss).

### 2.4. Confocal Microscopy

Confocal fluorescent images were taken using an inverted Zeiss LSM510 confocal microscope with 10× plain or 63× oil-immersion objective. Sequential excitation wavelengths at 488 nm and 543 nm were provided by argon and helium–neon gas lasers, respectively. Emission filters BP500-550 and LP560 were used for collecting green and red images in channels one and two, respectively. After sequential excitation, green and red fluorescent images of the same cell were saved and analyzed with ZEN software. The term colocalization refers to the coincident detection of above-background green and red fluorescence. The image size was set at 1024 × 1024 pixels.

### 2.5. RNA Isolation and Plasmid Constructs

Total RNA was isolated from cells harvested in a TRIzol reagent (Invitrogen) according to the manufacturer’s instructions. RNA was reverse-transcribed to cDNA using the M-MLV reverse transcriptional system (Invitrogen). Sequences encoding SLC17A9 were PCR-amplified by Pfu DNA polymerase (Agilent, Santa Clara, CA, USA) from cDNA derived from C2C12 cells and cloned into the pEGFP-N1 vector (Invitrogen). Primers used were SLC17A9-F (5′-TGCTCGAGCCATGCCATCCCAGCGCTC-3′) and SLC17A9-R (5′-GTGGATCCTCGAGGTCCTCATGAGTGG-3′). For knockdown studies in C2C12 cells, we constructed the plasmid pSUPER-SLC17A9. In order to do this, pSUPER plasmid (Ambion, Austin, TX, USA) was linearized using Bgl II and Hind III to facilitate directional cloning. Sequences for mouse SLC17A9-shRNA were as follows: #1 (5′-GATCCCCCCTTCCTGACATTCTCTCGAATTCAAGAGATTCGAGAGAATGTCAGGAAGGTTTTTA-3′) and #2 (5′-AGCTTAAAAACCTTCCTGACATTCTCTCGAATCTCTTGAATTCGAGAGAATGTCAGGAAGGGGG-3′).

For SLC7A9 stable knockdown studies, we constructed pLKO.1 vectors (8453, Addgene, Watertown, MA, USA) encoding shRNAs that target SLC17A9. Briefly, the plasmids were linearized using AgeI and EcroR1 to facilitate directional cloning. The sequences for human SLC17A9-shRNA are as follows: #1, 5′-CTTCAACCACAGTGGCATTTC-3′; #2, 5′-CACACTGTAGGATGCTTAAAG-3. Lentivirus was made using a three-plasmid packaging system. Briefly, pLKO vectors with SLC17A9 shRNAs were co-transfected with psPAX2 and pMD2.G into HEK293 T cells. Lentivirus was harvested 48 h after transfection, and 8 μg/mL polybrene was added. Sub-confluent Hela cells were infected with harvested lentivirus, and then were selected in 2 μg/mL puromycin for 3 days.

### 2.6. Western Blot

Proteins were analyzed using standard methods. Proteins derived from whole cells or lysosome lysates were resolved on 10% SDS-PAGE. Proteins were then transferred onto PVDF membranes using a semi-dry transfer apparatus (Bio-Rad, Milano, Italy). Non-specific binding was blocked using 5% skim milk in TBS-T (0.1% Tween-20 in 1× Tris-buffered saline, pH 7.4) for 2 h at room temperature. Membranes were then incubated with specific primary antibody solution at 4 °C overnight with gentle constant shaking. After three thorough TBS-T washes, the membranes were incubated with corresponding HRP-conjugated second antibody at room temperature for 1 h. Immunoreactive bands were visualized using Clarity ECL (Bio-Rad) and autoradiography.

### 2.7. Cell Viability Assay

Briefly, cells were transfected with indicated DNA using Lipofectamine 2000 or electroporation. Cell death was induced by treating cells with 5 μM DIDS for 24 h or silencing SLC17A9. Cells were stained with propidium iodide (P-4170, 10 μg/mL, Sigma, Oakville, Canada) in PBS for 15 min at room temperature. Cells were then immediately examined by confocal fluorescence microscopy. For quantitative analysis, adherent cells were collected by trypsinization and then mixed with detached cells in the culture medium, and PI^+^, GFP^+^ cells and GFP^+^/PI^+^ cells were counted using a hemocytometer under an upright fluorescence microscope.

For some assays, LDH activity was determined by measuring the NADH oxidation with pyruvate as the substrate. For each assay, a 50 μL sample was incubated with 50 μL reaction buffer (CytoTox-ONE™ Homogeneous Membrane Integrity Assay, Promega, Madison, WI, USA) for 10 min at 22 °C and the fluorescence at 340 nm was measured using a spectrophotometer. Results were scaled to complete cell death induced by exposure to 1% TritonX 100 at room temperature for 20 min. The percentage of cell death was defined as = 100 × (sample’s OD − background medium OD)/(TritonX 100 OD − medium background OD).

To determine the modalities of cell death, an Apoptosis/Necrosis Assay Kit (ab176749) was adopted to simultaneously monitor apoptotic and necrotic cells. The Apopxin Green (green) represented cell apoptosis through measuring the translocation of phosphatidylserine (PS). The 7-AAD (red) positive nucleus represented late-stage apoptosis and necrosis. Early-stage cell apoptosis (green), late-stage cell apoptosis (green and red) and necrosis (red) were simultaneously detected under a fluorescence microscope.

### 2.8. Data Analysis

Data are presented as mean ± SEM. Statistical comparisons were made using Student’s *t* test. *p* values of <0.05 were considered statistically significant. *: *p* < 0.05; **: *p* < 0.01.

## 3. Results

### 3.1. SLC17A9 Loss Associated Cell Death and Lysosome Dysfunction Are Rescued by TFEB Expression

SLC17A9 reportedly transports ATP into lysosomes as well as other secretory vesicles [5,10,28,29,30]. Recently, we showed that SLC17A9 deficiency resulted in cell death in both Cos1 and C2C12 cells [10]. The cell death in SLC19A9 deficient cells can result from dysfunction in secretory vesicles other than lysosomes. To exclude this possibility, we heterologously expressed TFEB-GFP [17,18,19], to rescue cell death induced by SLC17A9 deficiency. Cell death was monitored using propidium iodide (PI) staining. Previously, DIDS has been frequently used as SLC17A9 blocker [5,10]. Although it has also been shown to inhibit other transporters, our recent work [10] has indicated that DIDS directly inhibits SLC17A9-mediated ATP uptake in isolated lysosomes, and suppresses lysosomal ATP accumulation via SLC17A9 in intact cells. We have also shown that DIDS treatment impairs lysosome function, leading to cell death. Importantly, the effects of DIDS on ATP transport, lysosomal ATP accumulation, lysosome function, and cell viability are rescued by SLC17A9 overexpression [10]. Therefore, DIDS provides a reliable tool to suppress SLC17A9 and to study lysosomal ATP in lysosome function and cell physiology. In agreement with our previous findings, DIDS treatment resulted in 36.33 ± 2.96% PI positive Cos1 cells [10]. The percentage of PI positive cells induced by DIDS was reduced to 16.67 ± 2.19% by expressing TFEB-GFP but not lysosomal-associated membrane protein 1 (Lamp1)-GFP, which serves as a control (Figure 1A,B). We also assessed cell death by examining the levels of lactate dehydrogenase (LDH), a cytosolic enzyme that is released into cell culture media once cells lose membrane integrity. TFEB-GFP significantly decreased the percentage of cell death induced by DIDS from 49.67 ± 7.42% in the control (Lamp1-GFP) to 14.33 ± 2.85% (Figure 1C).

Previously, we have demonstrated that lysosomal ATP was also reduced by SLC17A9 shRNA but not scramble shRNA [10]. In agreement with our published data, SLC17A9 shRNA, but not scramble shRNA, significantly increased cell death in control C2C12 cells expressing Lamp1-GFP. As with TFEB’s effect on DIDS-induced cell death, TFEB-GFP expression dramatically decreased cell death induced by SLC17A9 shRNA in C2C12 cells. Although TFEB-GFP itself had no effect on cell viability, TFEB-GFP dramatically decreased PI positive cells induced by SLC17A9 shRNA from 38.2 ± 2.35% to 8.00 ± 1.95% (Figure 1D and Appendix A). LDH assay also revealed that TFEB-GFP expression markedly reduced cell death induced by SLC17A9 shRNA from 39.47 ± 1.85% to 8.60 ± 1.60% (Figure 1E). Because cell death induced by SLC17A9 inhibition or deletion was rescued by the master gene for lysosomal protein production, we suggest that lysosomes, but not other secretory vesicles, are the major organelles associated with SLC17A9 deficiency-induced cell death.

To determine the modalities of cell death, early-stage cell apoptosis (green), late-stage cell apoptosis (green and red) and necrosis (red) were simultaneously monitored using the Apoptosis/Necrosis Assay Kit [31]. As shown in Figure 1F,G, SLC17A9 deficiency induced both early-stage and late-stage apoptosis, but not necrosis.

Previously, we have also shown that the loss of mSLC17A9 compromises lysosomal function. To further test whether TFEB-GFP rescue of cell death is associated with lysosome function, DQ-BSA, an endocytic probe that emits fluorescence upon proteolytic activity of lysosomal Cathepsin D [32], was used. As we showed before [10], DIDS (5 μM) treatment reduced DQ-BSA signals in Lamp1-GFP expressing Cos1 cells which served as controls. This was corrected by the expression of TFEB-GFP (Figure 2A,B). In agreement with these data, in C2C12 cells, attenuated DQ-BSA staining induced by mSLC17A9 shRNA was rescued by the heterologous expression of TFEB-GFP (Figure 2C,D). The decreased DQ-BSA signals induced by SLC17A9 suppression could be due to a reduction in the number of lysosomes. To exclude this possibility, we measured Lamp1 expression. We did not observe a decrease in Lamp1 expression (Appendix A and Figure 3E,F) when SLC17A9 was inhibited, suggesting that the number of lysosomes was not altered by SLC17A9 suppression. Altogether, our data suggest that lysosomal SLC17A9 is critical for lysosomal function and cell viability.

### 3.2. SLC17A9 Deficiency Results in Cathepsin D Dysfunction

Cathepsin D is an aspartic protease that is involved in protein degradation in a strongly acidic milieu of lysosomes. Loss of Cathepsin D results in lysosome dysfunction, lysosomal storage, and cell death [12], the same phenotypes observed in SlC17A9 deficient cells [10]. Because ATP is suggested to directly activate Cathepsin D in a cell free measurement [16], and because SLC17A9 functions as a lysosomal ATP transporter, we suggest that SLC17A9 may play an essential role in regulation of proteolysis in lysosomes by maintaining the lysosomal ATP level that is required for Cathepsin D activation. To test this hypothesis, Cathepsin D activity in situ was detected by BODIPY FL-pepstatin A, a fluorescent probe that binds only to active Cathepsin D [33,34,35]. As shown in Figure 3A,B, strong BODIPY FL-pepstatin A signals were observed in Dextran–Texas Red loaded lysosomes of control Cos1 cells. DIDS (5 µM, 10 µM and 20 µM) dose-dependently eliminated BODIPY FL-pepstatin A signals. As expected, the loss of BODIPY FL-pepstatin A signals induced by 5 µM and 10 µM DIDS was completely and partially rescued by Cathepsin D-RFP [36] or SLC17A9-mCherry expression (Figure 3A,B), suggesting that DIDS suppresses Cathepsin D activity by inhibiting SLC17A9-mediated ATP transport. Interestingly, the loss of BODIPY FL-pepstatin A signals induced by 20 µM DIDS was not rescued by SLC17A9-mCherry expression (Figure 3A), likely due to an inhibition of exogenous SLC17A9 by a high dose of DIDS. The decreased Cathepsin D activity induced by DIDS could be due to a reduction in Cathepsin D expression. To exclude this possibility, the expression level of Cathepsin D was compared between control cells and cells treated with DIDS. As shown in Appendix A, Cathepsin D expression was not decreased by DIDS. These data suggest that DIDS inhibits Cathepsin D activity but not its expression.

Similarly, in C2C12 cells, SLC17A9 shRNA markedly reduced Cathepsin D activity reflected by the loss of BODIPY FL-pepstatin A signals (Figure 3C,D) but not Cathepsin D expression (Figure 3E,F). The loss of BODIPY FL-pepstatin A signals induced by SLC17A9 shRNA was rescued by Cathepsin D-RFP (Figure 3C,D). Because SLC17A9 shRNA did not significantly alter the expression of Cathepsin D, Cathepsin B, TFEB, and Lamp1 (Figure 3E,F), these data together suggest that impaired lysosome functions and cell death induced by SLC17A9 downregulation is likely attributed to a decrease in Cathepsin D activity but not the expression of Cathepsin D or other related proteins in lysosomes.

### 3.3. Cell Death and Lysosomal Dysfunction Induced by SLC17A9 Deficiency Are Rescued by Cathepsin D

To investigate whether cell death induced by SLC17A9 deficiency was mediated by defective Cathepsin D, Cathepsin D-GFP was co-transfected with SLC17A9 shRNA. Cathepsin D-GFP expression decreased the percentage of PI positive cells induced by SLC17A9 shRNA from 39.00 ± 4.46% in the control to 7.45 ± 0.61% (Figure 4A,B). LDH assay also showed a rescue effect of heterologous Cathepsin D on cell death induced by SLC17A9 shRNA. The percentage of cell deaths induced by SLC17A9 shRNA was reduced from 44.00 ± 6.28% in the control to 8.76 ± 1.70% by Cathepsin D-GFP expression (Figure 4C). The rescue effect of Cathepsin D-GFP was further investigated in cells treated with 5 µM DIDS. Cathepsin D-GFP expression dramatically reduced the percentage of PI positive cells from 41.67 ± 5.24% in control cells expressing Lamp1-GFP to 10 ± 3.61% (Figure 4D). Cell deaths measured using LDH assay also showed a significant reduction induced by 5 µM, from 45.67 ± 6.173% in the Lamp-GFP control group to 6.00 ± 2.08% in the Cathepsin D-GFP group (Figure 4E). In line with Figure 3A, Cathepsin D-GFP did not rescue the cell death induced by 20 µM DIDS (Figure 4D,E).

To further test the involvement of lysosomes in cell death induced by SLC17A9 deficiency, we monitored the effect of Cathepsin D-GFP on lysosomal function measured by DQ-BSA staining. Cathepsin D-GFP but not Lamp1-GFP rescued the de-staining of DQ-BSA induced by either DIDS (Figure 4F,G) or SLC17A9 shRNA (Figure 4H,I). Because low pH is essential for lysosomal functions, the loss of DQ-BSA signals caused by DIDS and SLC17A9 shRNA could be attributed to compromised lysosomal pH. To exclude this, lysosomal pH was monitored using LysoTracker Red DND-99, a red-fluorescent dye widely used to label acidic lysosomes [37,38,39]. As shown in Appendix A, neither DIDS nor SLC17A9 shRNA affected LysoTracker signals. Taken together, our data suggest that compromised SLC17A9 leads to dysfunctional Cathepsin D, subsequently resulting in lysosomal dysfunction and cell death.

### 3.4. SLC17A9 Deficiency Associated Cell Death Is Not Rescued by Cathepsin B

Cathepsins are a class of globular proteases in the lysosome. There are fifteen members of Cathepsin family, which are classified as serine proteases, cysteine proteases, or aspartic proteases based on their structure, catalytic mechanism, and their target proteins [40]. Cysteine protease cathepsin D (CTSD) and aspartic protease cathepsin B (CTSB) are the most abundant lysosomal proteases [8,41]. Cathepsin D belongs to aspartic proteases whereas Cathepsin B is a cysteine protease. To test whether cysteine proteases are also involved in cell death induced by SLC17A9 deficiency, we measured Cathepsin B activity using Magic Red-Cathepsin B that emits red fluorescence upon degradation by cathepsin B [34,35,42]. Neither DIDS (Figure 5A) nor SLC17A9 shRNA (Figure 5B) had any effect on Cathepsin B activity. Consistently, Cathepsin B expression did not rescue cell death induced by either DIDS (Figure 5C–E) or SLC17A9 shRNA (Figure 5F,G). These data suggest that aspartic proteases, such as Cathepsin B, are unlikely to be the key enzymes regulated by lysosomal ATP.

## 4. Discussion

Previous studies have indicated that SLC17A9 transports ATP into lysosomes as well as other secretory vesicles [5,8,10]. Deficiency in SLC17A9 causes cell death. However, the role of lysosomal SLC17A9 in cell viability and the molecular mechanism of cell death induced by SLC17A9 deficiency remain unclear. In this study, we showed that SLC17A9 deficiency-associated cell death was rescued by TFEB, a master gene for lysosomal protein production, suggesting that dysfunctional lysosomes are the major players in cell death induced by SLC17A9 deficiency.

Lysosome hydrolases are instrumental in degrading impaired organelles and macromolecules. Dysfunction of lysosome hydrolytic enzymes causes lysosomal storage, and subsequently leads to cell death [11]. Cathepsin D is one of the main hydrolases in lysosomes. Its deficiency has been associated with human lysosomal storage disorders [12,13,14]. However, the regulatory mechanism of Cathepsin D remains elusive with the exception of its acidic pH optimum. By using intact cells, we suggest that lysosomal ATP may be a critical regulator for Cathepsin D activation, identifying a molecular mechanism for Cathepsin D regulation. This is in agreement with a previous report using a cell free assay [16]. Although a decrease in lysosomal ATP by suppressing SLC17A9 results in compromised Cathepsin D, an aspartic protease, the lysosomal cysteine protease Cathepsin B was not affected by lysosomal SLC17A9 deficiency. Our results suggest that aspartic protease Cathepsin D might be the major hydrolytic enzyme that is regulated by ATP in lysosomes, although its regulatory mechanisms await further investigation. Since Cathepsin D deficiency is associated with human lysosomal storage disorders [12,13,14], our studies suggest that lysosomal ATP/SLC17A9 could be involved in lysosomal storage disorders.

## Figures and Tables

**Figure 1 cells-11-00887-f001:**
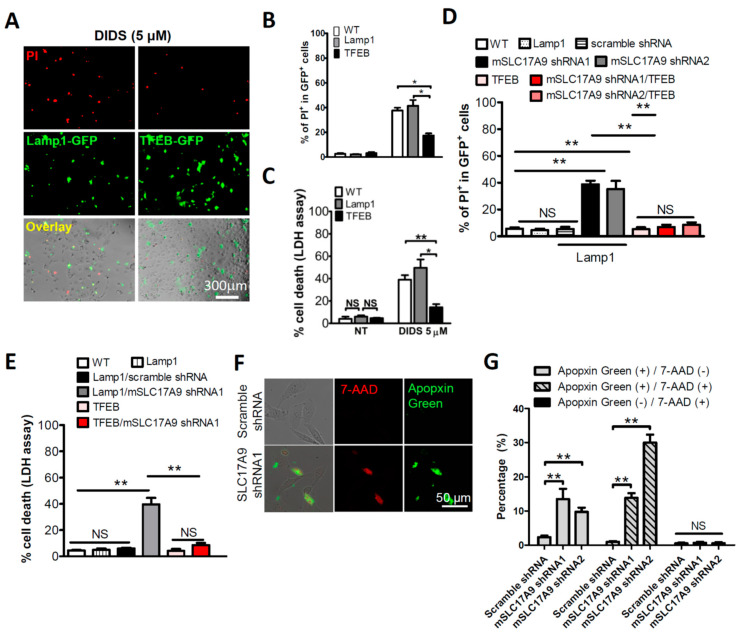
TFEB overexpression rescued apoptotic cell death induced by SLC17A9 deficiency. (**A**,**B**) TFEB overexpression rescued cell death induced by DIDS. Cos1 cells sham-transfected or transfected with Lamp1-GFP or TFEB-GFP were treated with 5 μM DIDS for 24 h. TFEB but not Lamp1 overexpression significantly reduced the percentage of PI^+^ cells. The percentage of PI^+^ cells in each experiment was calculated from ≥250 cells from randomly chosen fields, and the experiment was repeated independently 3 times in triplicate. (**C**) Percentage of cell death indicated by LDH assay showed that TFEB overexpression dramatically decreased cell death. The experiment was repeated 3 times in triplicate. (**D**,**E**) TFEB overexpression rescued SLC17A9 silence mediated cell death. C2C12 cells were transfected with Lamp1-GFP/SLC17A9 shRNA or TFEB-GFP/SLC17A9 shRNA using the Neon^®^ electroporation system. (**F**,**G**) The modalities of cell death were determined using an Apoptosis/Necrosis Assay Kit. SLC17A9 deficiency induced both early-stage (green) and late-stage (green and red) apoptosis but not necrosis (red). The experiment was repeated independently 3 times in triplicate and representative images are shown. WT: wild type (sham-transfection); NT: no treatment; NS: not significant. * *p* < 0.05; ** *p* < 0.01.

**Figure 2 cells-11-00887-f002:**
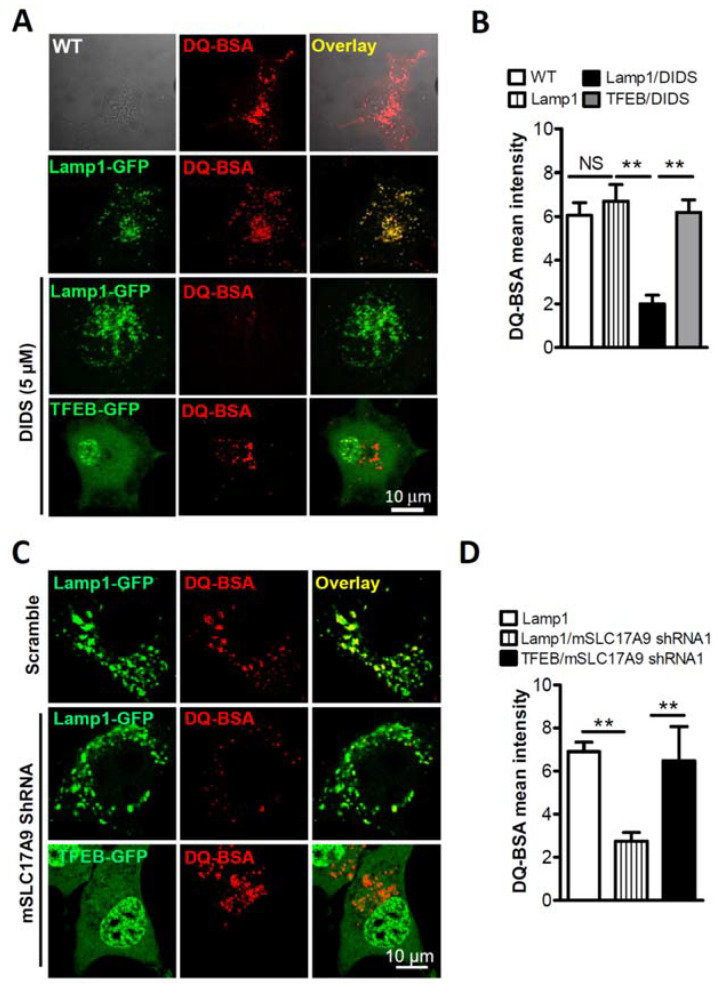
TFEB overexpression rescued the lysosomal dysfunction induced by SLC17A9 deficiency. (**A**,**B**) Decreased DQ-BSA signals in lysosomes of Cos1 cells treated by DIDS (5 μM, 6 h) were recovered by overexpressing TFEB but not Lamp1. The experiment was repeated independently 3 times in triplicate and representative images are shown. (**C**,**D**) Decreased DQ-BSA signals in lysosomes of mSLC17A9 knockdown C2C12 cells were recovered by overexpressing TFEB but not Lamp1. The experiment was repeated independently 3 times in triplicate and representative images are shown. WT: wild type (sham-transfection); NS: not significant. ** *p* < 0.01.

**Figure 3 cells-11-00887-f003:**
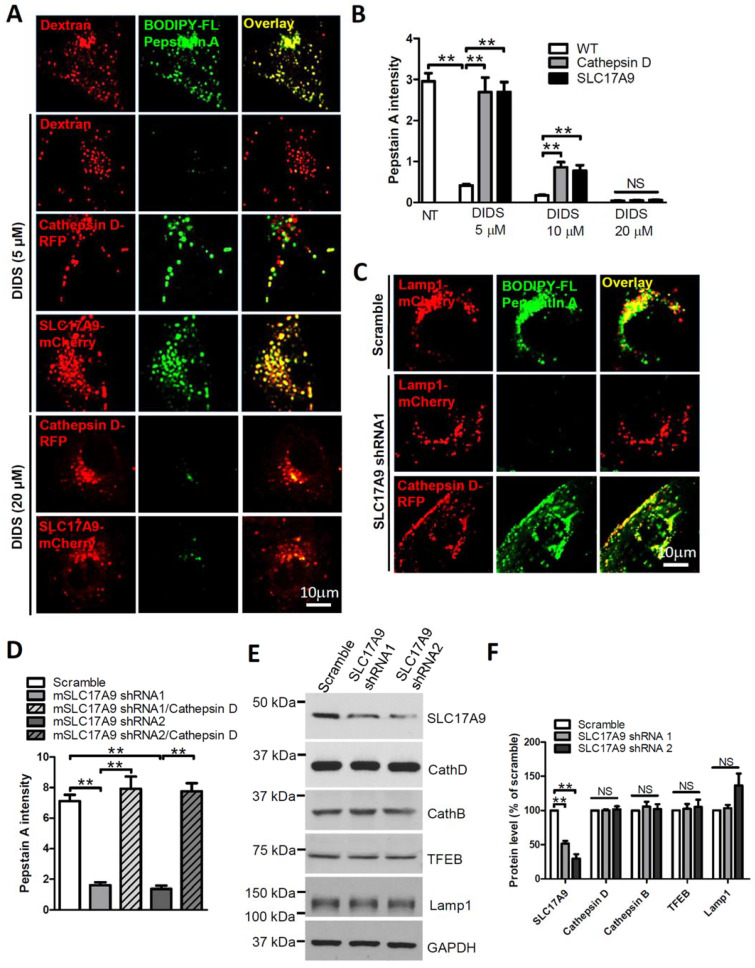
SLC17A9 deficiency impaired Cathepsin D activity in lysosomes. (**A**,**B**) DIDS treatment (5 μM, 6 h) significantly decreased the activity of Cathepsin D in lysosomes of WT Cos1 cells but not cells expressing Cathepsin D-RFP or SLC17A9-mCherry. Texas Red 10 kD Dextran was used to label lysosomes in WT Cos1 cells. BODIPY FL-pepstatin A (1 μM) was used to detect Cathepsin D activity in live cells. The experiment was repeated independently 3 times and representative images are shown. (**C**,**D**) Overexpressing Cathepsin D recovered decreased Cathepsin D activity in lysosomes of SLC17A9 deficient C2C12 cells. The experiment was repeated independently 3 times and representative images are shown. (**E**,**F**) The levels of Cathepsin D, Cathepsin B, TFEB, and Lamp1 were not altered by SLC17A9 knockdown in C2C12 cells. NT: no treatment; NS: not significant. ** *p* < 0.01.

**Figure 4 cells-11-00887-f004:**
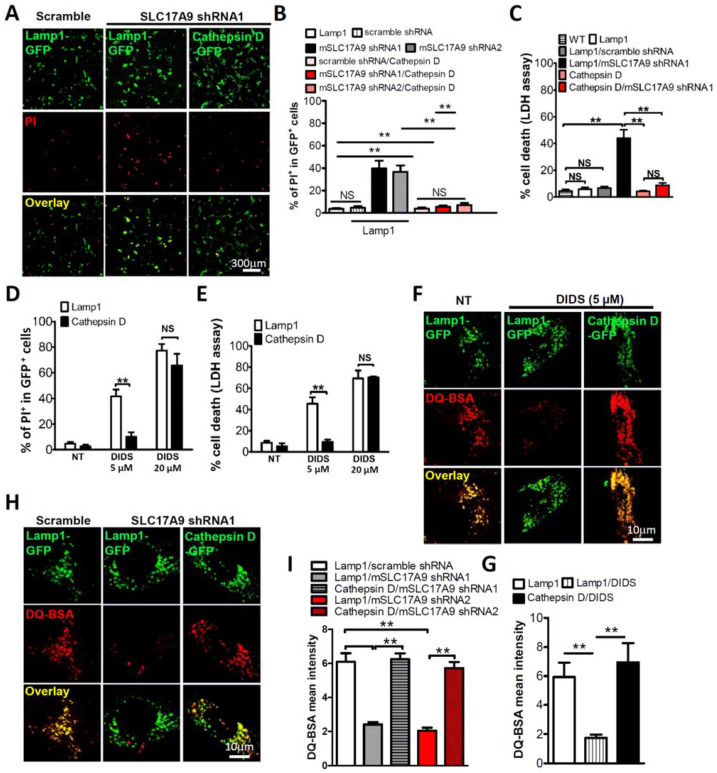
Cathepsin D overexpression rescued cell death induced by SLC17A9 deficiency. (**A**,**B**) Heterogeneous Cathepsin D rescued C2C12 cell from SLC17A9 knockdown associated cell death. The percentage of PI^+^ cells in each experiment was calculated from ≥250 cells from randomly chosen fields, and the experiment was repeated independently 3 times in triplicate. (**C**) Percentage of cell death indicated by LDH assay. The experiment was repeated 3 times in triplicate. (**D**) PI staining showing that Cathepsin D overexpression rescued the cell death induced by DIDS treatment in Cos1 cells. (**E**) LDH assay showing that Cathepsin D overexpression rescued the cell death induced by DIDS treatment in Cos1 cells. (**F**,**G**) Strong DQ-BSA signals in Cos1 cells overexpressing Cathepsin D but not Lamp1 upon DIDS treatment (5 μM, 6 h). The experiment was repeated 3 times in triplicate. (**H**,**I**) The decreased fluorescence of DQ-BSA in SLC17A9 knockdown C2C12 cells was recovered by Cathepsin D overexpression. The experiment was repeated independently 3 times in triplicate. WT: wild-type (sham-transfection); NT: no treatment; NS: not significant. ** *p* < 0.01.

**Figure 5 cells-11-00887-f005:**
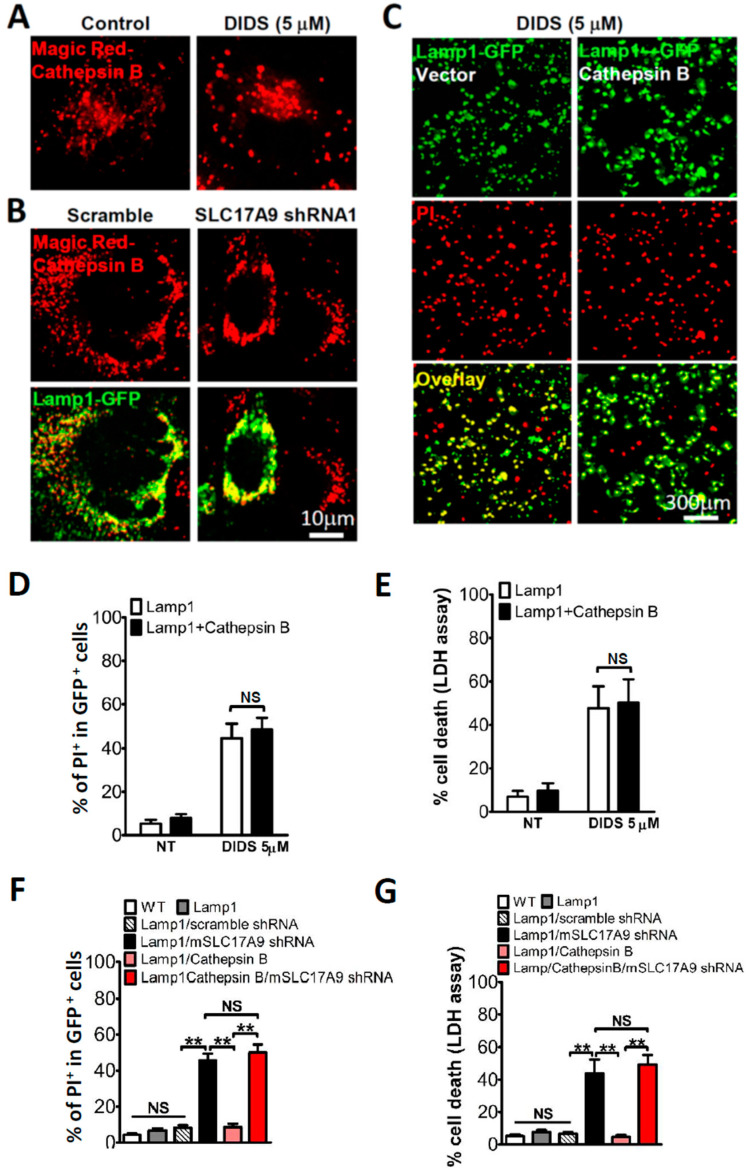
Cathepsin B activity was not affected by SLC17A9 deficiency. (**A**) DIDS (5 μM, 6 h) treatment did not alter lysosomal Cathepsin B activity in Cos1 cells. (**B**) Cathepsin B activity was not affected in SLC17A9 knockdown in C2C12 cells. (**C**–**E**) Overexpression of Cathepsin B did not rescue the cell death induced by DIDS treatment (5 μM, 24 h) in Cos1 cells assayed by PI staining (**C**,**D**) or LDH detection (**E**). Cells were transfected with Lamp1-GFP or co-transfected with Lamp1-GFP and hCathepsin B because hCathepsin B is not fluorescence-tagged. (**F**,**G**) Cathepsin B overexpression did not rescue C2C12 cell death induced by SLC17A9 knockdown. For PI staining in (**D**,**F**), the percentage of PI^+^ cells in each experiment was calculated from ≥250 cells from randomly chosen fields, and the experiment was repeated independently 3 times in triplicate. For LDH assay in (**E**,**G**), the experiment was repeated 3 times in triplicate. WT: wild type (sham-transfection); NT: no treatment; NS: not significant. ** *p* < 0.01.

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
