# Peer review of "Lysosomal ATP Transporter SLC17A9 Controls Cell Viability via Regulating Cathepsin D"

_cells, 2022, doi:10.3390/cells11050887_

Round 1

Reviewer 1 Report

It’s a straightforward experimental design with a clear outcome. Authors demonstrate that SLC17A9 deficiency causes cell death involved in the Cathepsin D pathway, not Cathepsin B.

Minor points

  1. line 39. It may be worth mentioning a few particular human diseases for readers.
  2. line 40. It may be worth specifying what kinds of phenotypes are the same.
  3. line 132. ANOVA with which test? Tukey?
  4. line 143. Reference 5 and 18 are the same article?
  5. fig1B. Is there a significant difference between WT and TFEB with DIDS? ANOVA with Turkey?
  6. fig1I. Miss labeled? Should it switch the upper labels of the black grid and straight striped grid?
  7. fig1F. Overlay of DIDS-treated TFEB-GFP + DQ-BSA showed the restored activity of ‘lysosomal cathepsin D’; however, this image showed DQ-BSA were not colocalized with TFEB, which doesn’t match to fig1H.
  8. line 203. Another clickable photoaffinity probe to identify cathepsin D (Zhul et al., Nat Commun 7, 13042 (2016)?

Author Response

Comments and Suggestions for Authors

It’s a straightforward experimental design with a clear outcome. Authors demonstrate that SLC17A9 deficiency causes cell death involved in the Cathepsin D pathway, not Cathepsin B.

Minor points

  1. line 39. It may be worth mentioning a few particular human diseases for readers.

Cathepsin D has been associated with lysosomal storage and cell death in mouse models and human disorders such as neuronal ceroid lipofuscinoses and neurodegeneration. This has been added to the new manuscript. Thanks.

  1. line 40. It may be worth specifying what kinds of phenotypes are the same.

The phenotypes have been specified, such as lysosomal dysfunction, lysosomal storage, and cell death.

  1. line 132. ANOVA with which test? Tukey?

Sorry, it was Student’s t-test. This has been corrected in the Methods.

  1. line 143. Reference 5 and 18 are the same article?

Thank you. This has been corrected.

  1. Is there a significant difference between WT and TFEB with DIDS? ANOVA with Turkey?

There is a difference between WT and TFEB with DIDS. This has been corrected. Thanks.

  1. Miss labeled? Should it switch the upper labels of the black grid and straight striped grid?

Corrected. Thanks.

  1. Overlay of DIDS-treated TFEB-GFP + DQ-BSA showed the restored activity of ‘lysosomal cathepsin D’; however, this image showed DQ-BSA were not colocalized with TFEB, which doesn’t match to fig1H.

Great point. TFEB is a cytosolic protein that translocates to the nucleus when the cell is in a stress condition such as starvation. It is not necessary that TFEB is colocalized with DQ-BSA. The reviewer’s question reminded us of whether DQ-BSA could be in the nucleus as shown in Figure 1H. After we reviewed our data, we found that DO-BSA was normally outside of the nuclei. To avoid any confusions, we replaced the images in Figure 1H (new Figure 2C).

  1. line 203. Another clickable photoaffinity probe to identify cathepsin D (Zhul et al., Nat Commun 7, 13042 (2016)?

Thanks for the great suggestion. Unfortunately, we were not able to perform the experiment because of the lack of the expertise with the technique. I guess it will take long time for me to train a student to perform the experiment. I hope that the review understands the situation.

Reviewer 2 Report

In this study, the authors revealed that cell death induced by SLC17A9 deficiency is rescued by the transcription factor EB (TFEB). They also found that Cathepsin D is inhibited by SLC17A9 deficiency while overexpression of Cathepsin D suppressed the cell death induced by SLC17A9 deficiency. Although the results are potentially interesting, I have several points that should be addressed by the authors. Specific comments are as follows.

Major points.

  1. The major concern with this manuscript is that the molecular mechanism of the inhibition of Cathepsin D by SLC17A9 deficiency is ambiguous. The expression level of Cathepsin D protein is unchanged, then what? Much more work is necessary to illustrate the mechanism in the manuscript.
  2. A further significant concern is that all the cells do not express SLC17A9. Is the cell death only induced in the SLC17A9-expressing cells? If that is the case, the specific cell name should be included in the title of the manuscript.
  3. The possible role of SLC17A9 should be discussed more in Introduction/Discussion.
  4. What is the cell death mode of SLC17A9 deficiency? Is that apoptosis? This point should be explained in the text.
  5. The authors should show the western blotting of SLC17A9 in order to verify the efficacy of shRNA in each figure. To check the complementation of phenotypes, the other related protein levels (Cathepsin D, B, TFEB, and Lamp1) should be examined by SLC17A9 shRNA.

Minor points.

  1. Figure 2A: There are errors in the concentration of DIDS.
  2. Also, a second siRNA is essential to rule out off-target effects.
  3. English should be carefully revised by a native English speaker or a professional editing service.

Author Response

Comments and Suggestions for Authors

In this study, the authors revealed that cell death induced by SLC17A9 deficiency is rescued by the transcription factor EB (TFEB). They also found that Cathepsin D is inhibited by SLC17A9 deficiency while overexpression of Cathepsin D suppressed the cell death induced by SLC17A9 deficiency. Although the results are potentially interesting, I have several points that should be addressed by the authors. Specific comments are as follows.

 Major points.

  1. The major concern with this manuscript is that the molecular mechanism of the inhibition of Cathepsin D by SLC17A9 deficiency is ambiguous. The expression level of Cathepsin D protein is unchanged, then what? Much more work is necessary to illustrate the mechanism in the manuscript.

In this study, we suggest that lysosomal dysfunction and cell death in SLC17A9 deficient cells are attributed to a reduction in Cathepsin D activity. However, a reduction in cathepsin D activity by SLC17A9 suppression could be due to a reduction in lysosomal ATP level and/or a decrease in cathepsin D expression. To distinguish the two possibilities, we measured the expression levels of cathepsin D. We found that cathepsin D was not changed by SLC17A9 knockdown. Together with our previous studies, we suggest that SLC17A9 transports ATP into lysosomes to activate Cathepsin D. Loss of SLC17A9 causes impaired lysosomes and cell death due to a reduction in Cathepsin D activity.

  1. A further significant concern is that all the cells do not express SLC17A9. Is the cell death only induced in the SLC17A9-expressing cells? If that is the case, the specific cell name should be included in the title of the manuscript.

I am not sure whether I understand the reviewer correctly. In fact, most cells tested express SLC17A9. For example, we previously showed that SLC17A9 is expressed in Cos1, HEK293, and C2C12 cells. Other groups have also suggested that SLC17A9 is a widely expressed protein (see doi: 10.1007/s11302-017-9568-1 and doi: 10.3389/fncel.2019.00224.). In this case, I think it is not necessary to include the specific cell type in the title of the manuscript. This may help attract more readers.

  1. The possible role of SLC17A9 should be discussed more in Introduction/Discussion.

As suggested, more information about SLC17A9 has been included in Introduction/Discussion.

  1. What is the cell death mode of SLC17A9 deficiency? Is that apoptosis? This point should be explained in the text.

Thanks for this great comment. We have suggested that SLC17A9 deficiency induces apoptosis but not necrosis by using the Apoptosis/Necrosis Assay Kit (Fig. 1F, 1G).

  1. The authors should show the western blotting of SLC17A9 in order to verify the efficacy of shRNA in each figure. To check the complementation of phenotypes, the other related protein levels (Cathepsin D, B, TFEB, and Lamp1) should be examined by SLC17A9 shRNA.

Good points. In fact, the efficacy of the SLC17A9 shRNAs used in this study has been demonstrated in our previous studies. In these studies, we showed the efficacy of the SLC17A9 shRNAs by using not only whole cell lysates but also in isolated lysosomes. Two papers have been properly cited in the manuscript (references9 and 10).

As the review suggested, we have further verified the efficacy of the two shRNAs in this study (Fig. 3E, 3F, S1). The expression of Cathepsin D, Cathepsin B, TFEB, and Lamp1 were also examined as the reviewer suggested (Fig. 3E, 3F).

 Minor points.

  1. Figure 2A: There are errors in the concentration of DIDS.

Corrected. Thanks.

  1. Also, a second siRNA is essential to rule out off-target effects.

Because both mSLC17A9 shRNA1 and mSLC17A9 shRNA2 showed similar effects on cell viability/PI staining (old Fig. 1D, 1E; 3A, 3B; new Fig. 1D, 1E; 4A, 4B), we decided to use only mSLC17A9 shRNA1 for DLH measurements and DQ-BSA experiments. As the reviewer suggested, we have used a second SLC17A9 shRNA to rule out off-target effects in this study. Please see Figure 1D-1G, 3D, 3F, 4B, 4I, 5F, and 5G.

  1. English should be carefully revised by a native English speaker or a professional editing service.

The manuscript has been proofread by a native English speaker in my lab.

Reviewer 3 Report

1)The introduction of this manuscript is far too brief for my taste. The author introduced the previous findings of SLC17A9 in cell death and Cathepsin D’s role in lysosome maintenance. While the author did not introduce any previous studies of the transcription factor EB(TFEB). The author should add some information about the research status of TFEB which will make section 3.1 more reasonable. Does the mutation of TFEB showing the similar cell death phenotype as the SLC17A9?

2) The figure 1D and 3B are kind of confusing. What does the x-axis “Lamp1” which covered all the bar except the WT mean? Based on the figure legend, it looks like the “Lamp1” beneath the x-axis is a mislabeling, please correct it.

3) In figure 1E and 1I(LDH assay and DQ-BSA assay), figure 3C and 3I, why there is only one mSLC17A9 shRNA data was shown? Could the author clarify the reason and mark which shRNA was used in these figure? Or the author could add both shRNA data.

4)In figure 2C and 2F, why did the author use the lysosome associated membrane protein 1(Lamp1) as a loading control in western? If inhibition of SLC17A9 induced lysosome dysfunction, then maybe Lamp1 should not be used as a control. The author should provide convincing statements, otherwise they should repeat the experiment with Actin or GAPDH as the loading control. At the same time, could the author also quantify the western blot band which will make the conclusion more trustable?

5)In section 3.2, the author claimed that the rescue failure of 20 μM DIDS by overexpressing SLC17A9-mCherry is due to the inhibition of exogenous SLC17A9 by a high dose of DIDS. Did the author try another relatively lower dose of DIDS, such as 10 μM? If so, how was the result? The author should add this result in the manuscript.

6) The author should explain or add some information in the methods sections to clarify why they did all the DIDS treatment experiments in Cos1 cells and all the shRNA related experiments in C2C12 cells.

7)Could the author explain why they co-transfected the Lamp1 and Cathepsin B in figure 4? It is unnecessary to co-transfect the Lamp1 and Cathepsin B. The author should clarify this, otherwise they should repeat the experiment by transfecting Cathepsin B only.

Minor points:

1)The DIDS concentration label in figure 2A is wrong. It should be 5 μM and 20 μM. Please correct it.

2) The author has already described the WT, NT and NS in the figure legend of figure 1, there is no need to repeat this in figure 3 and 4.

4) In figure 4F, the legend “Lamp1Cathepsin B/mSLC17A9 shRNA” should be “Lamp1/Cathepsin B/mSLC17A9 shRNA”.

Author Response

Comments and Suggestions for Authors

1) The introduction of this manuscript is far too brief for my taste. The author introduced the previous findings of SLC17A9 in cell death and Cathepsin D’s role in lysosome maintenance. While the author did not introduce any previous studies of the transcription factor EB(TFEB). The author should add some information about the research status of TFEB which will make section 3.1 more reasonable. Does the mutation of TFEB showing the similar cell death phenotype as the SLC17A9?

Great comments. More background information has been added to the Introduction and more data have also been collected to support our conclusions. As in the Introduction, mounting evidence suggests that ‘The transcription factor EB (TFEB) is a master regulator of lysosome biogenesis. … upregulation of TFEB increases lysosomal proteolysis, promotes lysosomal storage clearance, and protects cells from damage in lysosomal storage diseases.’

To test whether cells with TFEB deletion show the similar cell death phenotype as cells with SLC17A9 deficiency, we requested mTFEB-lentiCRISPRv2 from Dr. Eek-hoon Jho at the University of Seoul, Republic of Korea, and tried to generate TFEB KO C2C12 cells following his paper ‘Tan et al. TFEB regulates pluripotency transcriptional network in mouse embryonic stem cells independent of autophagy-lysosomal biogenesis.  Cell Death Dis. 2021 Apr 1;12(4):343. doi: 10.1038/s41419-021-03632-9’. Unfortunately, we did not successfully detect any TFEB reduction (mRNA and western blot) by the mTFEB-lentiCRISPRv2 after two attempts although it was showed that the mTFEB-lentiCRISPRv2 completely deleted TFEB in their cells.

The role of TFEB in cell viability has been studied before, and a large controversy has been generated, likely due to genetic compensation and experimental conditions (see papers listed below). Although TFEB KO created a large controversy in cell viability, the rescue effect of TFEB upregulation on impaired lysosomal proteolysis and lysosomal storage has been consistently reported (see references 20-27). In this case, we decide to not continue to make TFEB KO cells.

  • TFEB-KO/KD induces cell death:
  1. Slade et al. A lysosome independent role for TFEB in activating DNA repair and inhibiting apoptosis in breast cancer cells. Biochem J . 2020 Jan 17;477(1):137-160. doi: 10.1042/BCJ20190596.
  2. Sung et al. Inhibition of TFEB oligomerization by co-treatment of melatonin with vorinostat promotes the therapeutic sensitivity in glioblastoma and glioma stem cells. J Pineal Res . 2019 Apr;66(3):e12556. doi: 10.1111/jpi.12556. Epub 2019 Feb 14.
  3. Zhou et al. Transcription factor EB‑mediated autophagy promotes dermal fibroblast differentiation and collagen production by regulating endoplasmic reticulum stress and autophagy‑dependent secretion. Int J Mol Med . 2021 Feb;47(2):547-560. doi: 10.3892/ijmm.2020.4814. Epub 2020 Dec 9.
  4. Lu et al. Cyclodextrin Prevents Abdominal Aortic Aneurysm via Activation of Vascular Smooth Muscle Cell Transcription Factor EB. Circulation . 2020 Aug 4;142(5):483-498. doi: 10.1161/CIRCULATIONAHA.119.044803. Epub 2020 May 1.

  • TFEB-KO/KD does not induce cell death:
  1. TFEB, a master regulator of autophagy and biogenesis, unexpectedly promotes apoptosis in response to the cyclopentenone prostaglandin 15d-PGJ2. Acta Pharmacol Sin. 2021 Aug 20. doi: 10.1038/s41401-021-00711-7.
  2. Zhuang et al. Pharmacological enhancement of TFEB-mediated autophagy alleviated neuronal death in oxidative stress-induced Parkinson's disease models. Cell Death Dis . 2020 Feb 18;11(2):128. doi: 10.1038/s41419-020-2322-6.
  3. Yabuki et al. MiT/TFE family members suppress L-leucyl-L-leucine mythyl ester-induced cell death. J Toxicol Sci . 2021;46(3):143-156. doi: 10.2131/jts.46.143.
  4. Tan et al. Dihydromyricetin induced lncRNA MALAT1 - TFEB-dependent autophagic cell death in cutaneous squamous cell carcinoma. J Cancer . 2019 Jul 10;10(18):4245-4255. doi: 10.7150/jca.32807. eCollection 2019.
  5. Brady et al. The transcription factors TFE3 and TFEB amplify p53 dependent transcriptional programs in response to DNA damage. Elife . 2018 Dec 6;7:e40856. doi: 10.7554/eLife.40856.

  • TFEB-KO protects cells from cell death:
  1. Martina et al. TFEB and TFE3 are novel components of the integrated stress response. EMBO J . 2016 Mar 1;35(5):479-95. doi: 10.15252/embj.201593428. Epub 2016 Jan 25.

2) The figure 1D and 3B are kind of confusing. What does the x-axis “Lamp1” which covered all the bar except the WT mean? Based on the figure legend, it looks like the “Lamp1” beneath the x-axis is a mislabeling, please correct it.

Thank you for pointing these out. They are mislabeled and have been corrected.

3) In figure 1E and 1I(LDH assay and DQ-BSA assay), figure 3C and 3I, why there is only one mSLC17A9 shRNA data was shown? Could the author clarify the reason and mark which shRNA was used in these figure? Or the author could add both shRNA data.

Because both mSLC17A9 shRNA1 and mSLC17A9 shRNA2 showed similar effects on cell viability/PI staining (old Figures 1D, 1E, 3A, 3B, 4F, 4G; new Figures 1D, 1E, 4A, 4B, 5F, 5G), we decided to use only mSLC17A9 shRNA1 for DLH measurements and DQ-BSA experiments. As the reviewer suggested, we have used a second SLC17A9 shRNA to rule out off-target effects in this study. Please see Figure 1D, 1E, 1G, 3D, 3F, 4B, 4I, 5F, and 5G.

As the reviewer suggested, we have showed that both SLC17A9 shRNAs efficiently reduced SLC17A9 expression (Fig. 3E, 3F, and S1), but had no effect on the expression of Cathepsin D, Cathepsin B, TFEB, and Lamp1 (Fig. 3E, 3F). Both SLC17A9 shRNAs increased apoptosis (Figures 1D-1G), reduced Cathepsin D activity (Figures 3C, 3D), and compromised DQ-BSA signals (Figures 4H, 4I).

 4)In figure 2C and 2F, why did the author use the lysosome associated membrane protein 1(Lamp1) as a loading control in western? If inhibition of SLC17A9 induced lysosome dysfunction, then maybe Lamp1 should not be used as a control. The author should provide convincing statements, otherwise they should repeat the experiment with Actin or GAPDH as the loading control. At the same time, could the author also quantify the western blot band which will make the conclusion more trustable?

We showed that SLC17A9 suppression inhibited DQ-BSA signals. This could be due to a reduction in lysosome number. To exclude this possibility, we used Lamp1 as a control. We did not observe a decrease in Lamp1 level (Fig. 3E, 3F, S2), suggesting that the number of lysosomes was not altered by SLC17A9 suppression.

The decrease in Cathepsin D activity induced by suppressing SLC17A9 could be due to a reduction in Cathepsin D expression or other related proteins. To exclude this possibility, Cathepsin D expression level was compared. Our data showed that suppressing SLC17A9 did not reduce Cathepsin D expression and other related proteins, suggesting decrease in Cathepsin D activity induced by suppressing SLC17A9 is due to a reduction of luminal ATP but not the expression of Cathepsin D and other related proteins. As the review suggested, we have now used GAPDH as the loading control and the western blot bands were quantified (Fig. 3E, 3F).

 5)In section 3.2, the author claimed that the rescue failure of 20 μM DIDS by overexpressing SLC17A9-mCherry is due to the inhibition of exogenous SLC17A9 by a high dose of DIDS. Did the author try another relatively lower dose of DIDS, such as 10 μM? If so, how was the result? The author should add this result in the manuscript.

Great point. We have collected data suggesting that DIDS reduced BODIPY-FL Pepstatin A signals in a dose-dependent manner, and Cathepsin D-RFP and SLC17A9-mCherry rescued the decreased Pepstatin A signals induced by 5 μM DIDS (completely) and 10 μM DIDS (partially) but not 20 μM DIDS (Fig. 3A, 3B).

6) The author should explain or add some information in the methods sections to clarify why they did all the DIDS treatment experiments in Cos1 cells and all the shRNA related experiments in C2C12 cells.

We have been using Cos1 cell to study lysosomal membrane proteins because the cell has a spread-out morphology, making it easy to monitor proteins lysosomal localization. In the meantime, Cos1 cells are readily transfected. Thus, we used these cells for DIDS experiments as we did in two previous SLC17A9 studies (references 9 and 10). To assess the function of endogenous SLC17A9, we previously used pSUPER-mSLC17A9 shRNAs to knockdown mouse SLC17A9 in C2C12 (references 9 and 10).  Thus, we continued to use C2C12 cells for SLC17A9 knockdown experiments. To address question 4 from reviewer 2, Hela cells were used to generate SLC17A9 knockdown stable cell line to avoid the incompatibility with the kit (which needs both red and green channels for imaging) for cell death modality experiment because pSUPER-mSLC17A9 shRNAs have GFP. All these have been clarified in the Methods.

 7) Could the author explain why they co-transfected the Lamp1 and Cathepsin B in figure 4? It is unnecessary to co-transfect the Lamp1 and Cathepsin B. The author should clarify this, otherwise they should repeat the experiment by transfecting Cathepsin B only.

Because the Cathepsin B construct was not fluorescence-tagged, the lack of effect in the Cathepsin B group could be due to an unsuccessful expression of Cathepsin B. To avoid this, Lamp1-GFP was co-transfected to make sure that Cathepsin B was properly expressed in the cells. We have now specified in the figure legend.

 Minor points:

 1)The DIDS concentration label in figure 2A is wrong. It should be 5 μM and 20 μM. Please correct it.

Corrected. Thanks.

 2) The author has already described the WT, NT and NS in the figure legend of figure 1, there is no need to repeat this in figure 3 and 4.

Corrected. Thanks.

 4) In figure 4F, the legend “Lamp1Cathepsin B/mSLC17A9 shRNA” should be “Lamp1/Cathepsin B/mSLC17A9 shRNA”.

Corrected. Thanks.

Round 2

Reviewer 2 Report

The authors have made a great effort to address most of my concerns. I have no further points.